# Comparison of Loading Dose between Aflibercept and Faricimab for Neovascular Age-Related Macular Degeneration

**DOI:** 10.3390/jcm13020385

**Published:** 2024-01-10

**Authors:** Chikako Hara, Masaki Suzue, Satoko Fujimoto, Yoko Fukushima, Kaori Sayanagi, Kentaro Nishida, Kazuichi Maruyama, Shigeru Sato, Kohji Nishida

**Affiliations:** 1Department of Ophthalmology, Graduate School of Medicine, Osaka University, Suita 565-0871, Osaka, Japan; masaki.suzue@ophthal.med.osaka-u.ac.jp (M.S.); satoko.fujimoto@ophthal.med.osaka-u.ac.jp (S.F.); youko.fukushima@ophthal.med.osaka-u.ac.jp (Y.F.); kaori.sayanagi@ophthal.med.osaka-u.ac.jp (K.S.); nishiken@ophthal.med.osaka-u.ac.jp (K.N.); kazuichi.maruyama@ophthal.med.osaka-u.ac.jp (K.M.); s.sato@ophthal.med.osaka-u.ac.jp (S.S.); knishida@ophthal.med.osaka-u.ac.jp (K.N.); 2Department of Vision Informatics, Graduate School of Medicine, Osaka University, Suita 565-0871, Osaka, Japan; 3Integrated Frontier Research for Medical Science Division, Institute for Open and Transdisciplinary Research Initiatives (OTRI), Osaka University, Suita 565-0871, Osaka, Japan

**Keywords:** age-related macular degeneration, aflibercept, faricimab, anti-VEGF therapy, exudative change

## Abstract

Background: Recently, faricimab was approved as the new drug for neovascular age-related macular degeneration (nAMD). We lack the knowledge to choose between the existing drug and this new drug to use for treatment-naïve nAMD cases. In this study, we compared the functional and morphologic effects in loading dose between patients with treatment-naïve nAMD treated with either intravitreal aflibercept (IVA) or intravitreal faricimab (IVF) injection in a clinical setting. Method: This retrospective study included 30 eyes of 28 patients who started treatment with IVA between June and September 2022 and 30 eyes of 29 patients who were administered IVF between October 2022 and March 2023. All patients received three monthly IVA or IVF. The best corrected visual acuity (BCVA), central retinal thickness (CRT), and the proportion of eyes with residual exudative change at baseline and 1,2, and 3 months after initial treatment were compared between the groups. Results: The mean BCVA significantly improved from pre-treatment after the loading dose in the IVA group (0.46 ± 0.46–0.36 ± 0.37, *p* = 0.0047) but not in the IVF group (0.46 ± 0.41–0.44 ± 0.45, *p* = 0.60). The mean CRT significantly improved in both groups. The proportion of eyes with residual exudative change was greater in the IVF group than in the IVA group 2 months after the first treatment (*p* = 0.026). The analysis of cases that achieved complete resolution of exudative changes after the loading dose showed that the IVA group had a significant improvement in the BCVA, whereas the IVF group did not (*p* = 0.0047 and 0.20, respectively). Conclusions: Although both IVA and IVF significantly improved CRT, the BCVA improved significantly in the IVA group but not in the IVF group.

## 1. Introduction

Neovascular age-related macular degeneration (nAMD) is a progressive disease that causes central blindness [1]. Intravitreal injection of anti-vascular endothelial growth factor (anti-VEGF) drugs is currently the first-line treatment for nAMD [1,2,3]. To date, the formulations of ranibizumab [2], aflibercept [1], and brolucizmab [4,5] have been used to stabilize nAMD. Among them, aflibercept has been used extensively and is still the most widely used drug worldwide. Aflibercept is a recombinant fusion protein consisting of portions of VEGF receptor 1 and 2 extracellular domains fused to the Fc portion of human immunoglobulin G (IgG), which also blocks PlGF [6]. In 2022, faricimab was approved after the TENAYA and LUCERNE trials [7], and is a humanized, bispecific IgG monoclonal antibody that inhibits VEGF-A and angiopoietin-2 (Ang-2) [8]. Ang-2 expression increases in the vascular endothelium and is associated with inflammation or loss of pericytes in vessels [6]. These phase 3 trials demonstrated that the intravitreal faricimab injection was not inferior to the intravitreal aflibercept in the best corrected visual acuity (BCVA) and stabilization of neovascularization activity in patients with nAMD [7]. Recently, several studies of the effectiveness of faricimab for treatment-naïve nAMD in real-world practice have been published, and effects similar to those reported in the phase 3 clinical trial have been reported [9,10]. However, we lack the knowledge to choose between the existing drug and this new drug to use for treatment-naïve cases. This study compared the safety and efficacy of aflibercept and faricimab treatment for patients with nAMD in real-world settings.

## 2. Materials and Methods

The study was approved by the Ethics Committee of the Osaka University Graduate School of Medicine (approval number 10039) and followed the tenets of the Declaration of Helsinki. The need for informed consent was waived because of the retrospective nature of the research.

The study included a consecutive series of patients with treatment-naïve nAMD who visited Osaka University Hospital and started treatment with intravitreal aflibercept or faricimab injection. We excluded patients with myopia of >−6 diopters and a history of vitrectomy. The patients with first visits between June and September 2022 were treated with intravitreal aflibercept (IVA), and those with first visits between October 2022 and March 2023 were treated with intravitreal faricimab (IVF). Three monthly intravitreal injections were performed in all patients.

All patients underwent comprehensive ocular examination, including measurement of the BCVA using Landolt C charts, color fundus photography, and spectral-domain and swept-source optical coherence tomography (SD-OCT; Cirrus HD-OCT, Carl Zeiss Meditec Inc., Dublin, CA, USA, and SS-OCT; DRI-SS-OCT, Topcon Inc., Tokyo, Japan) at each follow-up visit.

Both OCT methods were used to evaluate intraretinal edema, subretinal fluid, and pigment epithelial detachment. Central retinal thickness (CRT) was defined as the distance between the internal limiting membrane and the presumed retinal pigment epithelium (RPE) at the fovea and was assessed using SD-OCT. Central choroidal thickness (CCT) was defined as the distance between the presumed RPE and chorioscleral interface at the fovea and evaluated using SS-OCT. Fluorescein angiography and indocyanine green angiography (ICGA) were performed to diagnose AMD, and its subtype was determined using a confocal scanning laser ophthalmoscope (HRA2; Heidelberg Engineering Inc., Dossenheim, Germany). 

The outcomes included BCVA, the resolution rate of exudative change (subretinal fluid (SRF), intraretinal fluid (IRF), and pigment epithelium detachment (PED)), CRT, and CCT at 1, 2, and 3 months after the first treatment. Subgroup analysis was also performed to compare the eyes with polypoidal choroidal vasculopathy (PCV) only. 

### Statistical Analysis

For the statistical analyses, the decimal visual acuity was converted to a logarithm of the minimum angle of resolution (logMAR) units. The clinical characteristics of the patients treated with IVA and IVF were compared with those not treated using the chi-squared test or Fisher’s exact test for non-numeric data and the Wilcoxon test for numeric data. A one-way analysis of variance was used to assess changes in the BCVA, CRT, and CCT in each group. The rate of eyes with residual exudative changes was compared between the IVA and IVF groups using the chi-squared test. The changes in the three parameters at each time point between the two groups were compared using the unpaired *t*-test. All statistical analyses were performed using JMP Pro version 17 software (SAS Institute Inc., Cary, NC, USA). A *p*-value < 0.05 was considered statistically significant.

## 3. Results

In total, 30 eyes of 28 patients treated with IVA (IVA group) and 30 eyes of 29 patients treated with IVF (IVF group) were included in this study. There were no cases in which the treatment was started but interrupted or switched during the 3 months. The demographic characteristics of the IVA and IVF groups are presented in Table 1. The two groups showed no significant differences in age, sex, or lesion size. Although no significant difference in the subtype of nAMD was observed, the rate of eyes with PCV tended to be smaller in the IVA group than those in the IVF group.

The mean logMAR BCVA at pre-treatment and 1, 2, and 3 months after the first treatment in the IVA group were 0.46 ± 0.46, 0.40 ± 0.42, 0.38 ± 0.40, and 0.36 ± 0.37, respectively (Figure 1A). Significant improvements were seen at 2 and 3 months (*p* = 0.095, 0.0088, and 0.0031 at 1, 2, and 3 months, respectively). In contrast, in the IVF group, the mean logMAR BCVA at pre-treatment and 1, 2, and 3 months after the first treatment were 0.46 ± 0.41, 0.45 ± 0.41, 0.44 ± 0.46, and 0.44 ± 0.45, respectively, and showed no significant improvement at all the points after the first treatment compared with pre-treatment (*p* = 0.89, 0.59, and 0.60 at 1, 2, and 3 months, respectively) (Figure 1A). The time course of the mean changes of logMAR BCVA from pre-treatment is shown in Figure 1B. A significant difference at 3 months was observed between both groups (*p* = 0.049).

The mean CCT significantly decreased from 205 ± 79 µm and 232 ± 88 µm at pre-treatment to 191 ± 92 µm and 208 ± 100 µm at 3 months in the IVA and IVF groups, respectively. The mean CRT at 1, 2, and 3 months after the first treatment significantly improved compared with pre-treatment in both groups (Figure 2A). No significant difference was observed in the changes in CRT at all the points (Figure 2B).

The rates of eyes with all types of residual exudative change in each group are shown in Figure 3A. In the IVA group, they were 0.47 (14/30), 0.20 (6/30), and 0.20 (6/30) at 1, 2, and 3 months after the first treatment, respectively. In contrast, in the IVF group, they were 0.70 (21/30), 0.47 (14/30), and 0.43 (13/30) at 1, 2, and 3 months after the first treatment, respectively. Significant differences were observed at 2 months between the two groups (*p* = 0.065, 0.026, and 0.050, respectively). The rates of eyes with IRF, SRF, and PED are shown in Figure 3B–D. No significant difference was observed in the rate of eyes with each type of exudative change.

Next, we investigated the cases that achieved complete resolution of exudative changes after the loading dose in both groups to evaluate whether residual exudative changes were the cause of the lack of visual improvement in the IVF group. Twenty-four eyes in the IVA group and seventeen eyes in the IVF group achieved complete resolution of exudative changes after the loading dose. In the IVA group, the logMAR BCVA significantly improved from 0.53 ± 0.48 at pre-treatment to 0.47 ± 0.44, 0.46 ± 0.41, and 0.42 ± 0.41 at 1, 2, and 3 months after the first treatment, respectively (Figure 4A). Significant improvements were observed at 2 and 3 months (*p* = 0.13, 0.035, and 0.0047 at 1, 2, and 3 months, respectively). However, in the IVF group, the logMAR BCVA was 0.50 ± 0.50, 0.50 ± 0.49, 0.49 ± 0.55, and 0.44 ± 0.53 at pre-treatment, 1, 2, and 3 months after the first treatment, respectively (Figure 4A). No significant improvement was observed (*p* = 0.84, 0.88, and 0.20 at 1, 2, and 3 months, respectively) (Figure 5). The time course of the mean changes of the logMAR BCVA from pre-treatment is shown in Figure 4B. No significant difference was observed between both groups.

In the IVF group, the CRT significantly decreased from 498 ± 227 µm at pre-treatment to 302 ± 146 µm, 229 ± 188 µm, and 217 ± 74 µm at 1, 2, and 3 months after the first treatment, respectively (*p* = 0.001, <0.0001, and <0.0001 compared with pre-treatment). In the IVA group, the CRT was 410 ± 257 µm, 314 ± 227 µm, 274 ± 171 µm, and 262 ± 152 µm at pre-treatment, 1, 2, and 3 months (*p* = 0.002, <0.0001, and <0.0001 at 1, 2, and 3 months compared with pre-treatment) (Figure 6A). In terms of changes in CRT, the IVF group was significantly greater than the IVA group at 2 and 3 months (*p* = 0.020 and 0.028, respectively) (Figure 6B). The rates of eyes with residual exudative changes were 0.38 (9/24) and 0.09 (2/29) in the IVA group and 0.44 (7/16) and 0.13 (2/16) in the IVF group at 1 and 2 months after the first treatment, respectively. No significant difference was observed between both groups.

The comparison of eyes with PCV alone is shown in Figure 7. In both groups, CRT significantly improved; however, only the IVA group showed a significant improvement in the mean logMAR BCVA (Figure 8). No significant difference was observed in the rate of eyes with residual exudative changes and eyes that achieved complete resolution of the polypoidal lesion between both groups.

An RPE tear occurred in one eye with type 1 macular neovascularization in the IVA group. One patient in the IVF group complained of vertigo after the start of treatment. There was no other ocular or systematic adverse event during the period.

## 4. Discussion

In this study, we compared the effects between IVA and IVF for eyes with treatment-naïve nAMD. The rate of eyes that achieved complete resolution of exudative change in the IVF group was significantly lower than that in the IVA group. A significant improvement in the mean BCVA was observed in the IVA group but not in the IVF group. Additionally, in the analysis of eyes that achieved complete resolution of exudative change, the mean logMAR BCVA did not improve in the IVF group; however, the change in CRT was more significant than in the IVA group.

Several previous studies have reported the effects of 3-month loading doses of IVA on eyes with AMD. The VIEW study, a large clinical trial, reported an improvement of +6–8 letters after the 3-month loading dose [3]. Other studies have also reported an average improvement of +5 letters, or 0.1 logMAR [11,12,13,14]. The results of the present study are comparable to these findings. Recently, there have been a few reports on the effect of loading doses of faricimab in real-world practice [9,11]. Matsumoto et al. reported that the logMAR BCVA significantly increased from 0.33 to 0.22 after the loading dose [9], and Mukai et al. reported that the logMAR BCVA significantly increased from 0.40 to 0.32 [11]. However, in this study, the BCVA showed no significant improvement in the IVF group. One possible reason for the difference in the change in the BCVA improvement in both groups was that the rates of eyes with residual exudative change after the loading dose in the IVF group were higher than in the IVA group [15,16,17,18]. In contrast, no significant improvement in the BCVA was observed, even in the eyes that achieved complete resolution of exudative change. This suggests that other factors, besides the higher rate of eyes with residual exudative changes, may be associated with the lack of visual improvement in the IVF group. In the analysis of the TENAYA and LUCERNE trials, the rate of eyes with complete resolution of exudative change in the IVF group was higher than or similar to that in the IVA group [7,19]. Several potential reasons may account for the differences observed in the results compared to those of the current study. These include the analysis being performed 8 weeks after four consecutive doses of faricimab and three consecutive doses of aflibercept, the only types of exudative changes evaluated were IRF and SRF, and the influence of racial differences (Asians made up less than 10% of the total population). In particular, the influence of racial differences is significant. All of the patients in this study were Japanese, and it cannot be ruled out that this fact may have influenced the faricimab-inferiority results in this setting. 

In this study, the mean CRT of all the eyes was similar in both groups; however, the IVF group had more cases of residual exudative changes. This was because the mean CRT of only the eyes that achieved complete resolution of exudative change in the IVF group was thinner than that of the IVA group. Interestingly, the mean reduction of CRT at 3 months with IVF was greater than that with IVA when only the eyes that achieved complete resolution of exudative change were considered. In the TENAYA and LUCERNE trials, faricimab tended to reduce CRT more than aflibercept. It is also possible that faricimab has a stronger ability to reduce CRT in effective cases [7,19,20]. 

The main limitations were this study’s retrospective and single-center nature and short-term outcomes. Furthermore, the number of patients was small, and all were Japanese. However, in this study, there were no cases in which treatment was changed midway due to inadequate efficacy, and all patients were examined. Although further prospective studies with larger sample sizes should involve long-term outcomes, this study is meaningful as a real-world short-term treatment outcome.

## 5. Conclusions

Both IVA and IVF treatments are effective in reducing exudation and retinal thickness. However, IVA was more likely to eliminate exudative changes and improve visual acuity in the short term.

## Figures and Tables

**Figure 1 jcm-13-00385-f001:**
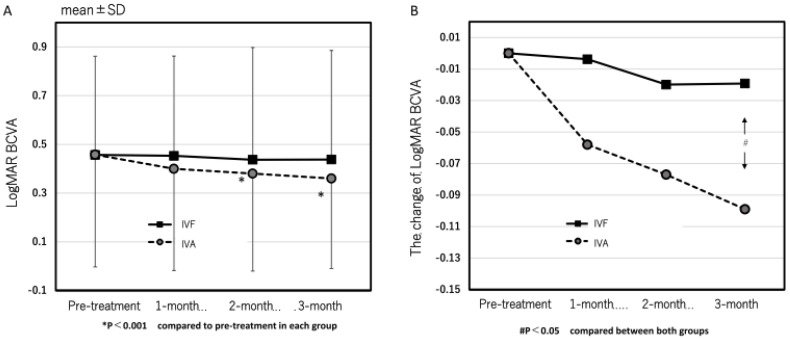
(**A**) The time course of best corrected visual acuity (BCVA) in both groups. In the IVA group, the BCVA significantly improved at 2 and 3 months after the first treatment (*p* = 0.0088 and 0.0031), but no significant improvement was observed in the IVF group. (**B**) The time course of change of the BCVA in both groups. A significant difference at 3 months was observed in both groups. (*p* = 0.049).

**Figure 2 jcm-13-00385-f002:**
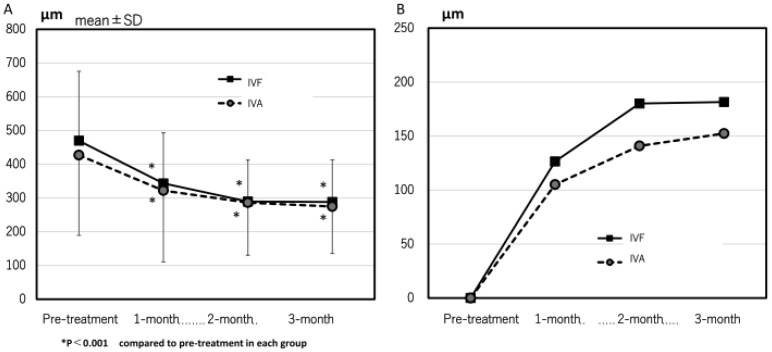
(**A**) The time course of central retinal thickness (CRT) in both groups. The mean CRT significantly decreased in both groups at all points. (**B**) The time course of the change in CRT in both groups.

**Figure 3 jcm-13-00385-f003:**
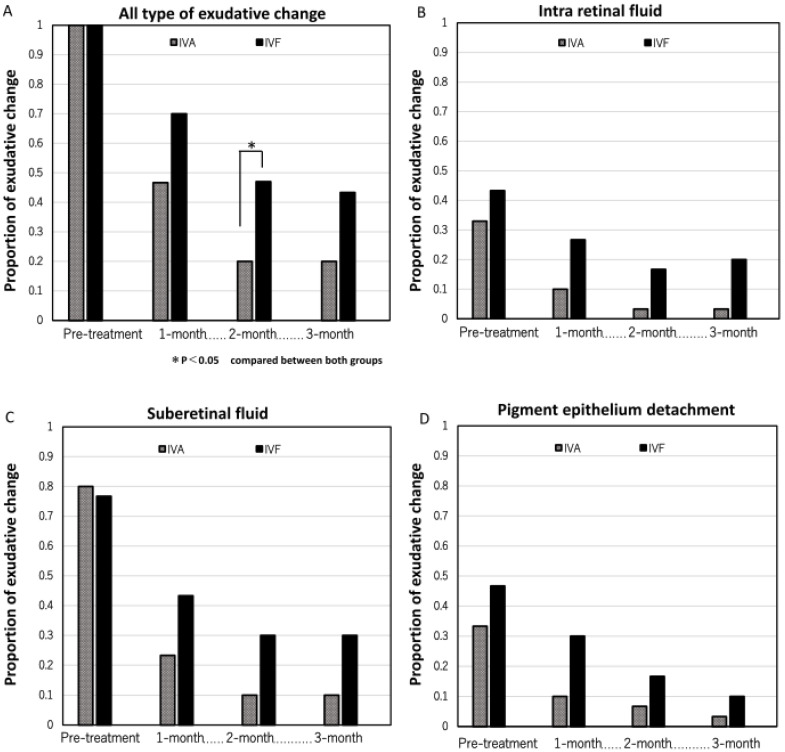
The proportion of eyes with residual exudative change in both groups. (**A**) The proportion of eyes with all types of exudative change (intraretinal fluid, subretinal fluid, and pigment epithelium detachment). The proportion of eyes with residual exudative change was greater in the IVF group than in the IVA group 2 months after the first treatment (*p* = 0.026). (**B**) The proportion of eyes with intraretinal fluid. (**C**) The proportion of eyes with subretinal fluid. (**D**) The proportion of eyes with pigment epithelium detachment.

**Figure 4 jcm-13-00385-f004:**
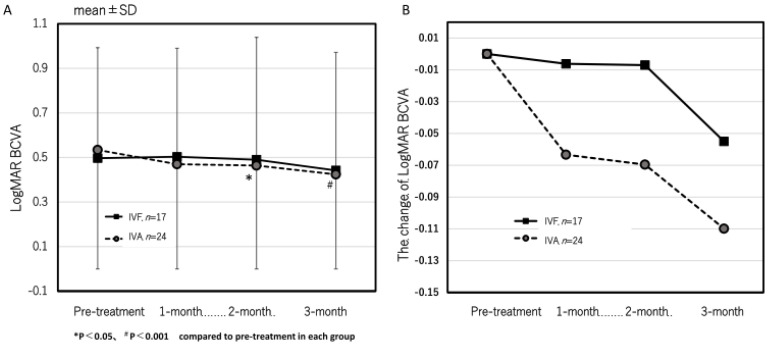
The time course of best corrected visual acuity (BCVA) (**A**) and change of BCVA (**B**) in eyes that achieved complete resolution of exudative change after the loading doses of both groups. In the IVA group, the BCVA significantly improved at 2 and 3 months after the first treatment (*p* = 0.035 and 0.0047), but no significant improvement was observed in the IVF group.

**Figure 5 jcm-13-00385-f005:**
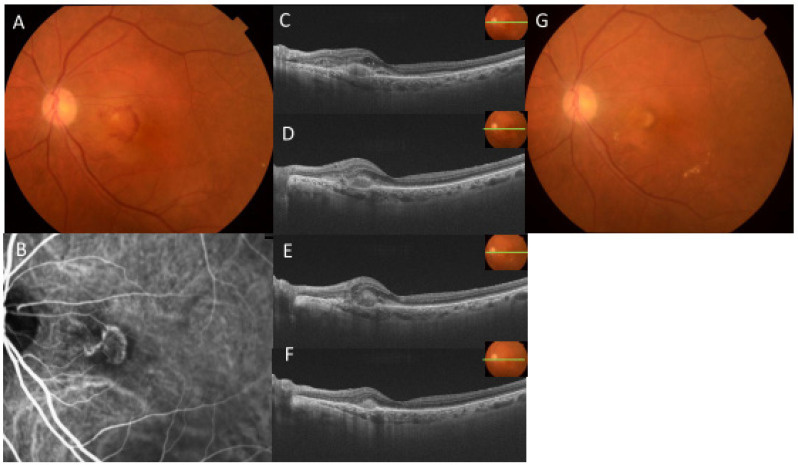
Representative case: 87-year-old female with type 2 macular neovascularization (MNV) treated with intravitreal injection of faricimab (IVF): The fundus photograph (**A**) and indocyanine green angiography (**B**) at pre-treatment showed retinal hemorrhage and MNV. Optical coherence tomography (OCT) (**C**) revealed type 2 MNV, intraretinal fluid (IRF), and subretinal fluid (SRF), and the best corrected visual acuity (BCVA) was 20/30. One month after the first IVF, OCT (**D**) revealed a resolution of IRF and SRF, but the BCVA deteriorated to 20/50. One month after the second IVF, OCT (**E**) revealed a little SRF, and the BCVA was 20/40. One month after the third IVF, OCT (**F**) showed complete resolution of SRF and IRF and a reduction of MNV, and fundus photograph (**G**) showed fibrotic scar. The BCVA was 20/40 and the improvement after IVF treatment was not observed. The BCVA was 20/40, and no improvement was observed after IVF treatment.

**Figure 6 jcm-13-00385-f006:**
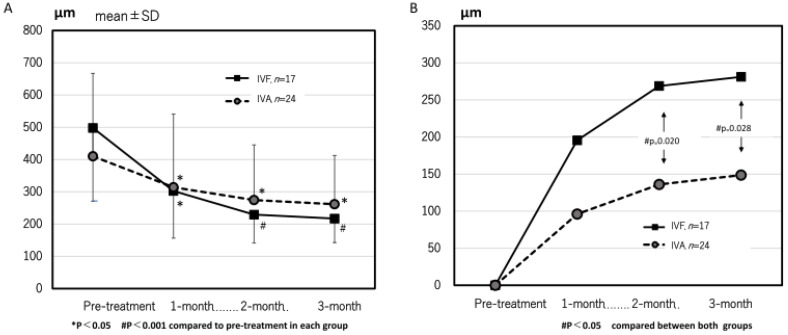
The time course of central retinal thickness (CRT) (**A**) and the change of CRT (**B**) in eyes that achieved complete resolution of exudative change after loading doses of both groups. The change in CRT in the IVF group at 2 and 3 months after the first treatment was significantly greater than in the IVA group. (*p* = 0.020 and 0.028, respectively).

**Figure 7 jcm-13-00385-f007:**
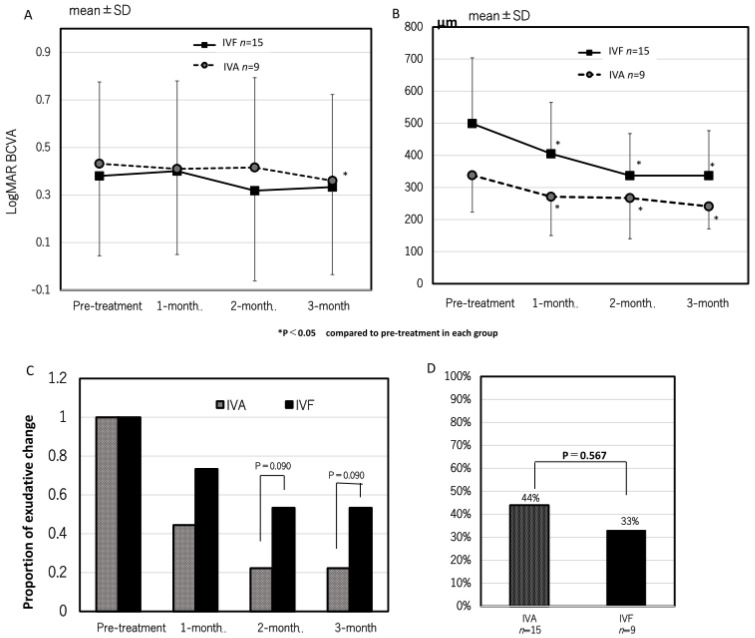
The results of eyes with polypoidal choroidal vasculopathy (PCV) only. The time course of best corrected visual acuity (BCVA) (**A**) and central retinal thickness (CRT) (**B**). In the IVA group, the BCVA significantly improved at 3 months after the first treatment (*p* = 0.010), but no significant improvement was observed in the IVF group. The mean CRT significantly decreased in both groups at all points. The proportion of eyes with residual exudative change (**C**) and eyes with complete resolution of polypoidal lesions (**D**) in both groups. There was no significant difference in these proportions.

**Figure 8 jcm-13-00385-f008:**
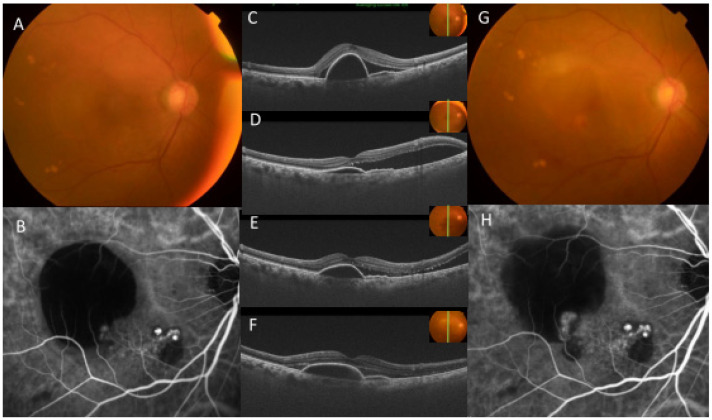
Representative case: 78-year-old female with polypoidal choroidal vasculopathy (PCV) treated with intravitreal injection of faricimab (IVF): The fundus photograph (**A**) and indocyanine green angiography (ICGA) (**B**) at pre-treatment showed pigment epithelium detachment (PED), polypoidal lesion, and branching network. Optical coherence tomography (OCT) (**C**) revealed PED and subretinal fluid (SRF), and the best corrected visual acuity (BCVA) was 20/32. One month after the first IVF (**D**), PED decreased, but SRF increased. The BCVA deteriorated to 20/62. One month after the second IVF (**E**), SRF and PED were revealed, and the BCVA was 20/62. One month after the third IVF, OCT (**F**) showed PED and SRF, and the fundus photograph (**G**) showed an increase in PED and materials like fibrin. ICGA (**H**) revealed aggravations of polypoidal lesions and a branching network. The BCVA was 20/62, and the deterioration of the BCVA after IVF treatment was observed.

**Table 1 jcm-13-00385-t001:** Baseline characteristics of all study eyes.

	IVF Group	IVA Group	*p*-Value
Number (eye/patient)	29/30	28/30	
Male/female	18/11	22/6	0.39
Age (mean ± SD, years)	79.4 ± 6.6	77.6 ± 9.6	0.38
AMD subtype (No. of eye)			0.31
Type 1 & 2 MNV	12	19	
Type 3	4	2	
PCV	15	9	
Baseline logMAR BCVA	0.46 ± 0.41	0.46 ± 0.46	0.99
Lesion size (GLD) (mean ± SD, µm)	2647 ± 1215	3247 ± 1579	0.12
Baseline CRT (mean ± SD, µm)	470 ± 206	427 ± 283	0.46
Baseline CCT (mean ± SD, µm)	191 ± 92 µm	205 ± 79	0.22

IVF; intravitreal faricimab, IVA; intravitreal aflibercept, SD; standard deviation, MNV; macular neovascularization, logMAR; logarithm of the minimum angle of resolution, BCVA; best corrected visual acuity, PCV; polypoidal choroidal vasculopathy, GLD; greatest liner dimension, CRT; central retinal thickness, CCT; central choroidal thickness.

## Data Availability

All data generated or analyzed during this study are included in this published article.

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
