# Peer review of "Comparison of Loading Dose between Aflibercept and Faricimab for Neovascular Age-Related Macular Degeneration"

_jcm, 2024, doi:10.3390/jcm13020385_

Round 1
Reviewer 1 Report
Comments and Suggestions for Authors
I would like to congratulate the authors. A few issues that need attention are the following:
1. Why compare aflibercept and faricimab response after 3 months while we know that faricimab requires 3 intravitreal injection for the loading dose?
2. Do you think that the PCV percentages in both groups could influence the results? Could you present subgroup analysis for the pCV eye only?
3. Regarding the safety of the two agents, did you identify and PED tears in your follow up or other complications?
Author Response
- Why compare aflibercept and faricimab response after 3 months while we know that faricimab requires 3 intravitreal injection for the loading dose?
Thank you for the question. As you pointed out, in the phase 3 clinical trial, the loading dose of faricimab is four injections. However, no standard specifies four instead of three, and the package insert also states three or four times for the induction phase. Therefore, to make a fair comparison with aflibercept, the present study compared treatment outcomes after three injections.
- Do you think that the PCV percentages in both groups could influence the results? Could you present subgroup analysis for the pCV eye only?
Thank you for the suggestion. Accordingly, we have added the analysis of eyes with PCV only (Fig. 7). Because the comparison of eyes with PCV alone was similar to the comparison of all cases, we believe that the difference in the percentage of PCV eyes had little effect on the results.
Line 194–198: The comparison of eyes with PCV alone is shown in Fig. 7. In both groups, CRT significantly improved; however, only the IVA group showed a significant improvement in mean logMAR BCVA (Fig. 8). No significant difference was observed in the rate of eyes with residual exudative changes and eyes that achieved complete resolution of the polypoidal lesion between both groups.
- Regarding the safety of the two agents, did you identify and PED tears in your follow up or other complications?
Thank you for your question. We have included the information in the last paragraph of the result section as follows:
Line 230–232: An RPE tear occurred in one eye with type-1 macular neovascularization in the IVA group. One patient in the IVF group complained of vertigo after the start of treatment. There was no other ocular or systematic adverse event during the period.
Reviewer 2 Report
Comments and Suggestions for Authors
Faricimab, as a novel treatment for macular edema, has shown promising clinical trial results and is being promoted, but the results in the real world are yet to be confirmed. This paper compared the efficacy of faricimab and aflibercept, including changes in BCVA and CRT. It was found that faricimab did not significantly improve BCVA, but could better reduce CRT, and the results were real and reliable. The main problems of the paper are as follows: 1, it is recommended to use the currently widely accepted visual acuity indicator, the ETDR visual acuity table (letters) for comparison. 2. The author compared the residual exudative change of the two drugs, but the residual exudative change caused by faricimab was more obvious, which was more significant with the reduction of CRT, which seemed to be contradictory. Please use specific pictures of original cases, and use representative pictures to show more convincing. 3. It is difficult to provide only three months 'loading dose data to explain the problem. Most experiments are one-year treatment data. If not, patients should have at least six months of treatment and follow-up.
Author Response
1, it is recommended to use the currently widely accepted visual acuity indicator, the ETDR visual acuity table (letters) for comparison.
Thank you for this comment. Although I agree with your indication, performing the visual acuity test on all patients using the ETDRS visual acuity table during our daily outpatient practice where many patients must be tested is difficult. Because it requires a lot of time per patient However, ETDRS visual acuity and decimal visual acuity are related, and we believe we have captured the visual acuity trends.
- The author compared the residual exudative change of the two drugs, but the residual exudative change caused by faricimab was more obvious, which was more significant with the reduction of CRT, which seemed to be contradictory. Please use specific pictures of original cases, and use representative pictures to show more convincing.
Thank you for your advice. The time course of the mean CRT of all eyes in both groups was similar; however, more patients had residual exudative changes in the faricimab group. In eyes with residual exudative change, the CRT is thicker. However, the mean CRT in only eyes with complete resolution of exudative change in the faricimab group was lower than that in the aflibercept group. Therefore, the mean CRT in all eyes was similar. This information was added to the discussion, as shown below. We also added representative cases as Figs. 6 and 8.
Line 273-276: In this study, the mean CRT of all eyes was similar in both groups; however, the IVF group had more cases of residual exudative changes. This was because the mean CRT of only the eyes that achieved complete resolution of exudative change in the IVF group was thinner than that of the IVA group.
- It is difficult to provide only three months 'loading dose data to explain the problem. Most experiments are one-year treatment data. If not, patients should have at least six months of treatment and follow-up.
Thank you for this comment. As you pointed out, we think that the outcome of the loading phase alone is not sufficient as a treatment outcome. However, there have been some real-world treatment results of faricimab for treatment-naïve AMD (Matsumoto et al. Greafe Arch Clin Ophthalmol 2023; doi: 10.1007/s00417-023-06116-y., Mukai et al. Sci Rep 2023; doi: 10.1038/s41598-023-35759-4., Inoue et al. Greafe Arch Clin Ophthalmol 2023; doi: 10.1007/s00417-023-06241-8.), all of which reported short-term results for loading dose only. In addition, many options are available for treating patients with AMD, and drugs can be immediately switched after starting treatment in poor responders. As shown in the present study, we will switch to other anti-VEGF drugs for cases with residual exudative changes after the loading dose in real-world practice. Therefore, cases with poor responses would be dropped out, and comparisons of long-term treatment results would not yield representative results. Therefore, in this study, we evaluated short-term treatment results without dropout cases.
Round 2
Reviewer 1 Report
Comments and Suggestions for Authors
No further queries raised.
Author Response
We appreciate your suggestions and supports.
Reviewer 2 Report
Comments and Suggestions for Authors
The author made the necessary response to my comments, obviously the author did not seriously respond to my ETDR visual chart suggestion. At present, the main problems are as follows: 1. The data on visual acuity and retinal thickness in all figures are not professional, and the standard deviation should be marked, although the standard deviation is very large. 2. The information content of each figures is small, and the arrangement of the figures is very arbitrary, such as Figure 5 and Figure 8, only several color photos, fluorescence fundus angiography and OCT images of two patients, why use two figures, and can't you synthesize one figure? Most importantly, the scan lines of OCT also need to be marked in the color map, otherwise how can they be compared? 3, as a high-level journal submission, the author should carefully edit the figures and tables.
Author Response
The author made the necessary response to my comments, obviously the author did not seriously respond to my ETDR visual chart suggestion.
We are very sorry that we are unable to respond to your suggestion within this study, as we do not perform the visual acuity test using ETDRS chart.
- The data on visual acuity and retinal thickness in all figures are not professional, and the standard deviation should be marked, although the standard deviation is very large.
Thank you for your suggestion. We added error bar of standard deviation in Fig 1,2,4, 6 and 7.
- The information content of each figures is small, and the arrangement of the figures is very arbitrary, such as Figure 5 and Figure 8, only several color photos, fluorescence fundus angiography and OCT images of two patients, why use two figures, and can't you synthesize one figure? Most importantly, the scan lines of OCT also need to be marked in the color map, otherwise how can they be compared? 3, as a high-level journal submission, the author should carefully edit the figures and tables.
Thank you for your suggestion. We added color figures with the scan lines in Figure 5 and 8.
Since Figure 5 and Figure 8 are two different cases, we have divided them into two separate Figures because we feel that combining them into one Figure would be confusing for the reader.